# Global sociodemographic, clinical, and epidemiological profiling of patients with mycetoma: A systematic review

**Mohamed Elmuntasir Salah**[1]*, **Michelle L. Fearon Scales**[2], **Kirlus Habib**[3], **Fadila Alhamwi**[1], **Suad Abdelwahab**[4], **Yassin Ahmed**[5], **Manal Mohamed Khalid**[6,7], **Dallas J. Smith**[2], **Ahmed Fahal**[1]

**1** Mycetoma Research Center, Khartoum, Sudan, **2** Mycotic Diseases Branch, Centers for Disease Control and Prevention, Atlanta, Georgia, United States of America, **3** Omdurman Teaching Hospital, Khartoum, Sudan, **4** Mohamed El-Amin Hamad Hospital, Khartoum, Sudan, **5** Princess Alexandra Hospital, Harlow, United Kingdom, **6** College of Medicine, Almughtaribeen University, Khartoum, Sudan, **7** Faculty of Medicine, Sudan University of Science and Technology, Khartoum, Sudan

* mohamed.elmuntasir.salah@gmail.com

## Abstract

### Background

Mycetoma is a neglected tropical disease that affects subcutaneous tissues. This chronic granulomatous inflammatory disease often leads to high morbidity rates, including amputation, disability and social stigma. Despite its substantial impact, the epidemiology of mycetoma remains largely unknown. This systematic review aimed to establish a global sociodemographic and clinical profile of affected patients and characterise the geographic patterns of the causative organisms of mycetoma.

### Methods

The study followed the 'Preferred Reporting Items for Systematic Review and Meta-Analysis' (PRISMA). The search strategy covered all key databases without restriction on language, setting, or year of publication. All observational studies in which the mycetoma patients' sociodemographic profile was described were included. Study quality was evaluated using a modified Newcastle-Ottawa Quality Assessment Scale. We calculated the mean percentage of patients reporting sociodemographic and clinical characteristics. We also determined the geographic patterns of the identified causative organism for reported mycetoma patients based on actinomycetoma and eumycetoma aetiology and genus-level taxonomic classifications. The included studies were heterogeneous in terms of population source, data collection method, and reported data from demographic characteristics to outcomes. This, in turn, can be due to the absence of a standardised data reporting form for

**Data availability statement:** All data and code used in this study are provided in a public repository hosted by https://github.com/CDCgov and registered with a DOI https://doi.org/10.5281/zenodo.15684055 and an Apache-2.0 license at the following URL https://github.com/CDCgov/mycetoma-systematic-review-2024.

**Funding:** The author(s) received no specific funding for this work.

**Competing interests:** The authors have declared that no competing interests exist.

mycetoma, which has limited the data analysis and our ability to compare patient characteristics and disease epidemiology over time and between regions.

## Results

Of 16,564 studies identified, 72 met the inclusion criteria, covering 35,004 persons, of which 29,328 were patients with mycetoma. We found that most cases originated from Sudan, India, and Mexico. The disease primarily affected males (74%) and rural residents (73%), with farmers being the most common occupation. Most patients were adults aged between 20 and 50 years (mean 36.2 years), and the lower limb was affected most (77%). Thirty-three percent of patients were cured on treatment, 15% had amputations, and 18% experienced recurrence. Eumycetoma was predominantly identified in Africa and the Arabian Peninsula, and actinomycetoma was more common in India and Mexico. The most common causative species were *Madurella mycetomatis* and *Actinomadura madurae*.

## Conclusion

This review provides a current understanding of the sociodemographic and clinical characteristics of patients with mycetoma worldwide. Most affected cases were adult males and rural residents, with the lower limb involvement being the most common. The distribution of causative organisms varied by region. The variability in outcomes and organisms underscores the complexity of the disease, highlighting the need for further research to understand its global impact.

### Author summary

Mycetoma is a neglected, devastating disease that frequently affects populations in tropical and subtropical regions. This infectious disease can be caused by a wide range of fungi, causing eumycetoma, and bacteria, causing actinomycetoma. Mycetoma can cause severe disability if not appropriately treated. Despite this substantial impact on patient health and community well-being, the epidemiological characteristics of the disease are still poorly understood. In this systematic review, we included 72 studies that cover 29,328 mycetoma cases, aiming to characterise the sociodemographic patterns, clinical features, and geographic distribution of the causative organisms. We found that mycetoma predominantly affects males residing in rural areas, especially farmers, and that the lower limb is the most affected body part. The outcome of these cases varied; some were cured, experienced recurrence or underwent amputation, which reflects the complexity of the disease treatment. We also observed that eumycetoma was reported more commonly in Africa, and actinomycetoma was more prevalent in India and Mexico. This systematic review provides a foundation for future research and contributes to a better understanding of the disease epidemiology.

## Introduction

Mycetoma is a chronic, disabling subcutaneous granulomatous inflammatory disease caused by a variety of fungi (eumycetoma) or bacteria (actinomycetoma) [1,2]. Without treatment, the disease progressively infiltrates deep tissues and bones, resulting in deformities and can be fatal [1,3]. The clinical characteristics of the disease include swelling, multiple sinuses, and grain-containing purulent or seropurulent discharge [1,4]. Although the foot is the most commonly affected area, other body parts can be affected [5]. Mycetoma primarily affects the tropical and subtropical regions between the latitudes of 15° south and 30° north of the equator, known as the 'mycetoma belt' [6,7]. This belt includes Sudan, Somalia, Senegal, India, Yemen, Mexico, Venezuela, Colombia, Argentina, and other countries [6,7]. The distribution of causative agents of mycetoma varies globally; eumycetoma and actinomycetoma both cause substantial disease in Africa, Asia, and Latin America. Even within a country, the distribution of the organisms may vary [8,9].

In 2016, the World Health Organisation recognised mycetoma as one of the neglected tropical diseases (NTD) [10]. The disease, particularly in early manifestations, exhibits similar features to other skin NTDs, leading to misdiagnosis and delays in appropriate treatment. It can lead to lifelong disabilities, reducing quality of life and imposing a burden on family members [11]. Mycetoma predominantly affects populations in remote regions, and patients often experience social stigma [12]. Although mycetoma is a public health problem with a high morbidity rate, the epidemiological characteristics of the disease are not fully understood [3,6]. A literature review by Van de Sande (2013) identified 8,763 cases of mycetoma reported globally [8]. This estimate likely underrepresents the true burden of disease as routine surveillance is lacking in most countries.

This study aimed to describe patterns in the existing literature to capture the sociodemographic and clinical profile of patients with mycetoma. We also explored global patterns in the causative organisms of mycetoma, classified by actinomycetoma and eumycetoma and by genus. This review provides a comprehensive synthesis of the key sociodemographic and epidemiologic characteristics among mycetoma patients and identifies regionally causative organisms of mycetoma across the world to improve diagnosis, identification of the causative species, and selection of an appropriate treatment for mycetoma.

## Methods

The systematic review was performed and reported according to the Preferred Reporting Items for Systematic Reviews and Meta-Analysis (PRISMA) group guidelines [13]. The study protocol is registered with PROSPERO under the registration number CRD42023451462.

### Search strategy

A comprehensive search was conducted in electronic databases including PubMed, Scopus, Web of Science, OVID, VHL, and Google Scholar to identify relevant publications. Grey literature on the occurrence of mycetoma and the sociodemographic characteristics of mycetoma patients was also searched. Additionally, a manual search of the reference lists of related papers was performed. No restrictions were placed on language, publication date, or country. Search strategies have been listed in the PRISMA flowchart (Fig 1). A detailed search strategy, including all search terms, is available in S1 Table.

### Review method and eligibility criteria

All observational studies reporting the occurrence of mycetoma were considered eligible if patients' sociodemographic features were described. Exclusion criteria were studies that reported secondary data (systematic review, literature review), studies that did not have clear methodology, studies limited to certain groups, case reports, case series, duplicated articles, and abstracts. We excluded case reports as they often focus on unusual cases, which may not accurately

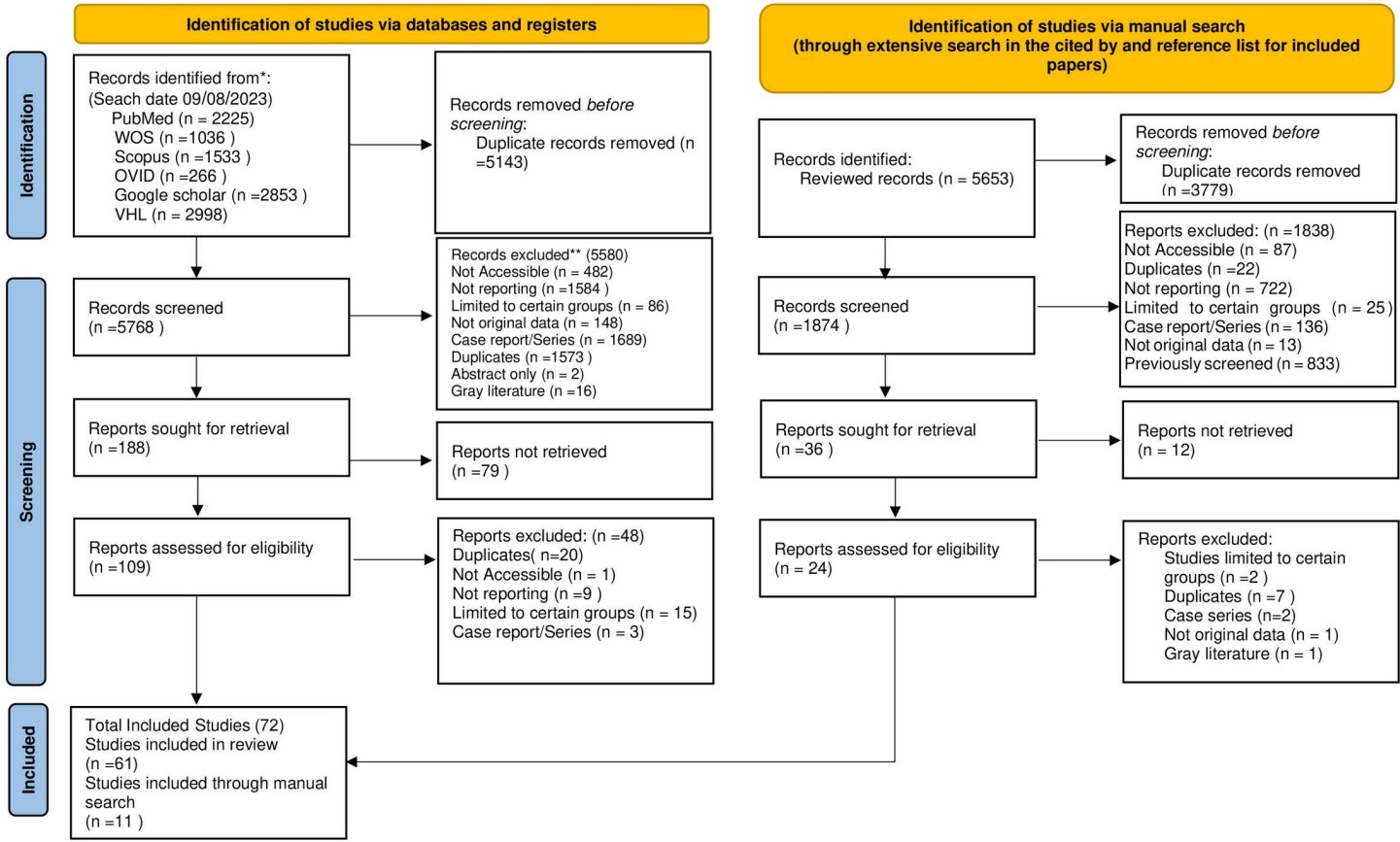

**Fig 1. PRISMA flowchart for mycetoma profiling.**

reflect the broader population of patients with mycetoma and introduce publication bias. In contrast, observational studies provide a more representative overview and reduce selection bias.

The initial screening was conducted independently by four authors based on predefined criteria, and then discrepancies were reviewed and resolved in pairs. Discrepancies between pairs were resolved by mutual agreement or by a third author. The full text screening of articles eligible for inclusion were reviewed independently by four reviewers. Discrepancies were resolved through consensus. Studies originally written in languages other than English were translated using Google translation service.

## Data extraction

Using the Cochrane library modified pre-piloted extraction form, two researchers independently extracted the following data from the eligible studies: Study description (first author, year of publication, country, total number of mycetoma patients, total study participants size, study design, study population source, and diagnostic methods), sociodemographic profile of patients (mean age, sex, educational level, residence, and occupation), and clinical characteristics of patients (comorbidities, disease severity, involved body regions, disease outcomes, and treatment received). Additionally, we recorded the number of patients with mycetoma for whom the causative organism was identified to species, genus, or aetiology. When information was unclear or missing, a consensus with a third reviewer was reached.

## Quality assessment of studies

To examine the quality of the included studies, a modified Newcastle-Ottawa Quality Assessment Scale was utilised by two independent investigators [14]. The scale assessed the study quality across three domains: cohort selection, which considers the sample's representativeness; comparability, which examines the control of confounding factors and the similarity of study groups; and outcome, which evaluates blinding, follow-up duration, and the validity of statistical tests. The overall score ranges from 0 to 9, with studies scoring 0–4 as poor quality, 5–6 as fair quality, and 7–8 (7–9 in the case of cohort and case control studies) as high quality. In the event of any discrepancies, a consensus was reached between the two investigators, with a third reviewer acting as an arbiter if necessary. Low-quality studies were included in our analysis to prevent data loss and provide valuable insights. However, their low scores were mainly due to methodological details rather than validity, and their publication in peer-reviewed journals adds credibility.

## Statistical analysis

Data analysis was conducted using R version 4.4.0. We produced descriptive statistics for each sociodemographic and clinical characteristic reported in the included studies. For studies that reported the mean age, we calculated the mean and range. For all other categorical variables, we first calculated the percentage of mycetoma patients within each subcategory for each study by dividing the reported number of patients for each group by the total number of mycetoma patients included in the study. Then we calculated the mean percentage and range for all studies that reported on a given subcategory. For some sociodemographic and clinical characteristics, studies did not report on all of the same sets of possible subcategories; therefore, the number of studies included in the mean percent calculation may differ among subcategories. Additionally, some studies reported on more detailed sociodemographic and clinical subcategories than others, so we had to reclassify to broader groupings that could be consistent across all the studies. For education level, we considered individuals with primary education and higher as literate and those without education or only Khalwa (traditional religious education often limited to Qur'anic studies) as likely illiterate. For occupation, we reduced the reported occupations to 12 broader categories, including animal breeder, farmer, freelancer (e.g., trader, merchant, shopkeeper, fishermen, etc.), government/desk job (police officer, teacher, military soldier, clerk, priest, etc.), housewife, non-skilful worker (maid, housekeeper, etc.), professional (engineer, veterinarian, mechanic, accountant, etc.), skilful worker (carpenter, bricklayer, blacksmith, gardener, driver, etc.), student, unemployed, and not reported. For the involved body region, we recategorized all of the detailed reported body parts into five body regions (buttocks & groin, face & neck, lower limbs, trunk, and upper limbs) where patients could be reported multiple times among these categories but only counted once within each individual category, along with two other broad categories for multiple sites reported or unspecified body site involvement. Out of the 72 articles reviewed [11,15–84], three articles were excluded from the demographic or clinical characteristics analyses due to being unable to differentiate mycetoma patient characteristics from the rest of the study population [70,76] or the study was not an epidemiological study but did include causative organism data [45]. All three of these studies were included in the causative organism analysis described below.

We also evaluated whether there were any geographic patterns in causative organisms for mycetoma patients at two different scales: identification to aetiology (e.g., actinomycetoma vs. eumycetoma) and genus (n = 58 studies). For the analysis at the aetiology level, we calculated the proportion of patients across studies that had mycetoma causative organisms identified as actinomycetoma, eumycetoma, unidentified, or not reported for each country. For the analysis at the genus level, we similarly calculated the proportion of patients across studies that had mycetoma-causative organisms identified to each genus for each country. However, the genus-level analysis excluded patients that had no reported causative organism or unidentified causative organisms at higher taxonomic levels (e.g., identified to aetiology, but not to genus-level). For this reason, Ethiopia and the United Arab Emirates were not included in the genus-level analysis since studies from these countries did not have any isolates identified beyond the actinomycetoma and eumycetoma aetiologies.

## Results

### Study eligibility results

Our systematic review identified 16,564 potentially relevant studies through database searches and manual examination of citation and reference lists. After removing duplicates, 7,641 studies were screened by title and abstract and 7,418 were excluded for the following reasons: inaccessibility (n = 569), not reporting relevant data (n = 2306), focusing on certain groups only (n = 111), not presenting original data (n = 161), being case reports/series (n = 1,825), duplicates (n = 1,595), abstracts only (n = 2), grey literature (n = 16), and 833 studies identified during manual search that had been screened previously during database screening conducted before the manual search screening. Of the remaining 223 studies, 91 studies were not retrieved. The full texts of the remaining 132 articles were assessed, and 56 articles were excluded for not meeting the eligibility criteria. Finally, 72 studies were included in this systematic review [11,15–84] (Fig 1).

### Quality assessment results

Detailed quality ratings of study quality were conducted according to the modified Newcastle-Ottawa criteria. Thirty-two studies were deemed to have an overall high-quality score, two studies had a fair quality, and 38 studies were considered low-quality (Fig 2 and S2 Table). Studies that were considered low-quality generally had low scores in the cohort selection and comparability domains, while the outcome domain was of high quality across all included studies.

The summary plot demonstrates the quality rating of the 72 included studies across three domains, cohort selection, comparability and outcome, and the overall quality judgement. The colour-coded ratings indicate the quality within each domain and the overall quality: green (high quality), yellow (fair quality), red (low quality). Individual quality scores for each study are listed in S2 Table, and the graph was generated using robavis package [85].

### Characteristics of included studies

Across all of the included studies, there were 35,004 persons and 29,328 patients with mycetoma; the study population (i.e., the denominator for the study) and number of mycetoma patients included in the studies varied widely (Table 1). The study population ranged from 11 to 6,983 patients, with a mean of 486.2 patients. The number of patients with mycetoma included in the studies ranged from 5 to 6,983 persons, with a mean of 407.3 patients. The geographical distribution of

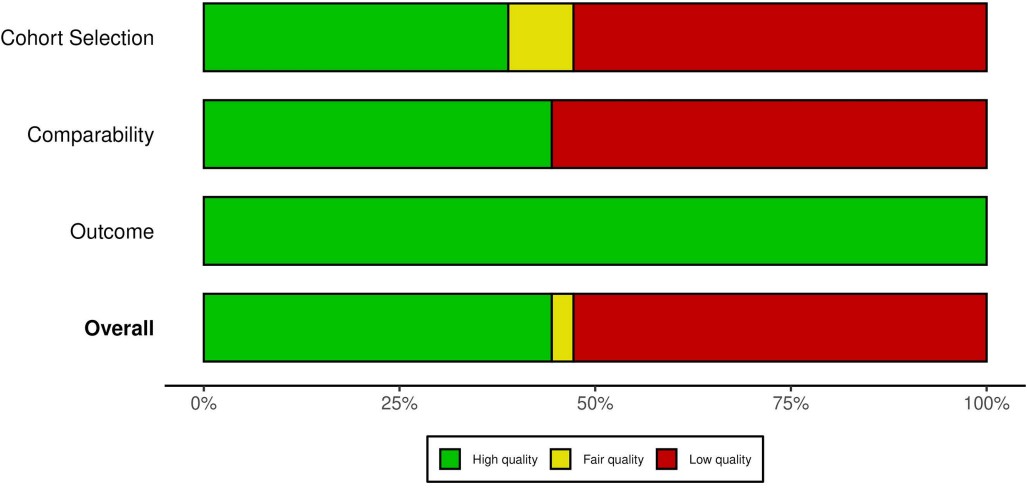

**Fig 2. Quality assessment of the included studies using the modified Newcastle-Ottawa criteria.**

Table 1. Summary of the main characteristics of the 72 included studies.

| ID | Authors | Country | Total Mycetoma patients | Total study population | Study design | Study Population Source | Diagnosis tools | Age | Gender | Education | Residency | Occupation | Comorbidities | Severe disease | Body region | Outcomes | Treatment | Causative Organism | Citation |
|---|---|---|---|---|---|---|---|---|---|---|---|---|---|---|---|---|---|---|---|
| ID 5105 | Abbas et al. 2018 | Sudan | 300 | 300 | Not mentioned | Single-center, hospital-based study | clinical, microscopic examination and cytology | X | X | | | X | | | X | X | | | 12 |
| ID 4075 | Abd El Bagi et al. 2003 | Sudan | 516 | 516 | Not mentioned | Single-center, hospital-based study | clinical, imaging, cytology and culture | | X | | | | | | | | | | 65 |
| ID 1249 | Aboudi et al. 2017 | Sudan | 100 | 100 | Prospective observational | Single-center, hospital-based study | Not mentioned | | X | | | X | | X | X | X | X | | 31 |
| ID 5785 | Adoubryn et al. 2008 | Côte d'Ivoire | 87 | 149 | Retrospective observational | Single-center, hospital-based study | Not mentioned | | X | | | | | | X | | | X | 85 |
| ID 3734 | Adoubryn et al. 2009 | West Africa | 46 | 85 | Retrospective observational | Multi-center, hospital-based study | clinical, culture, microscopic examination | | X | | | X | | X | X | | | X | 58 |
| ID 2267 | Ali et al. 2022 | Sudan | 160 | 160 | Cross-sectional | Community-based study | Not mentioned | | X | | | | | | X | | | | 41 |
| ID 5774 | Ansari et al. 2023 | India | 38 | 38 | Retrospective observational | Single-center, hospital-based study | clinical, imaging, microscopic examination, culture and genetics | X | X | | | | | X | X | | | X | 80 |
| ID 801 | Aounallah et al. 2023 | Tunisia | 18 | 18 | Retrospective observational | Single-center, hospital-based study | clinical, imaging, culture, and microscopic examination | | X | | | X | | | X | X | X | X | 27 |
| ID 475 | Azraga et al. 2011 | Sudan | 26 | 229 | Cross-sectional | Community-based study | Not mentioned | | X | X | X | X | | | X | | | | 23 |
| ID 1892 | Bakhiet et al. 2018 | Sudan | 1032 | 1032 | Not mentioned | Single-center, hospital-based study | clinical and microscopic examination | | X | | | X | | | | | | X | 39 |
| ID 4018 | Balabanoff et al. 1998 | Bulgaria | 18 | 18 | Not mentioned | Clinical-based | clinical, imaging, microscopic examination and culture | | X | | | | | | X | | | X | 63 |
| ID 5273 | Basher et al. 2021 | Sudan | 50 | 95 | Case-control | Single-center, hospital-based study | clinical, genetic, culture and microscopic examination | X | X | | | | | | | | | | 72 |

(Continued)

Table 1. (Continued)

| ID | Authors | Country | Total Mycetoma patients | Total study population | Study design | Study Population Source | Diagnosis tools | Age | Gender | Education | Residency | Occupation | Comorbidities | Severe disease | Body region | Outcomes | Treatment | Causative Organism | Citation |
|---|---|---|---|---|---|---|---|---|---|---|---|---|---|---|---|---|---|---|---|
| ID 1203 | Batalla et al. 2011 | Morocco | 12 | 12 | Retrospective observational | Single-center, hospital-based study | clinical, imaging, culture and microscopic examination | | X | | | X | | | X | X | X | X | 30 |
| ID 770 | Bocarro et al. 1893 | India | 100 | 100 | Not mentioned | Single-center, hospital-based study | Not mentioned | | X | | | | | | X | | | | 26 |
| ID 3905 | Bonifaz et al. 2014 | Mexico | 482 | 482 | Cross-sectional | Single-center, hospital-based study | clinical, microscopic examination and culture | | X | | | | | | X | | | X | 61 |
| ID 3081 | Castro et al. 1992 | Brazil | 41 | 41 | Not mentioned | Single-center, hospital-based study | clinical, culture, histopathology | | X | | | X | | X | X | X | X | X | 48 |
| ID 5603 | Castro et al. 2008 | Brazil | 27 | 27 | Not mentioned | Single-center, hospital-based study | clinical, imaging, culture and microscopic examination | | X | | | X | | X | X | X | X | X | 75 |
| ID 5771 | Colom et al. 2023 | Kenya | 58 | 60 | Cross-sectional | Community-based study | clinical, microscopic examination, culture and genetics | | X | | | | | | | | | X | 78 |
| ID 136 | Convit et al. 1959 | Venezuela | 37 | 146 | Retrospective observational | Multi-center, hospital-based study | Not mentioned | | X | | | X | | | X | | | X | 18 |
| ID 2893 | Correa et al. 2018 | Mexico | 174 | 174 | Cross-sectional | Single-center, hospital-based study | clinical, culture, microscopic examination | | X | | | X | X | | X | | | X | 45 |
| ID 5797 | Czechowski et al. 2001 | United Arab Emirates | 20 | 20 | Retrospective observational | Single-center, hospital-based study | clinical, histology, imaging | X | X | | X | | | | X | | | X | 87 |
| ID 3927 | Daoud et al. 2005 | Tunisia | 13 | 13 | Retrospective observational | Single-center, hospital-based study | clinical, imaging, microscopic examination, culture | | X | | | | | X | X | X | X | X | 62 |
| ID 3486 | Darrè et al. 2018 | Togo | 33 | 33 | Retrospective observational | Single-center, hospital-based study | clinical, microscopic examination | X | X | | | X | | X | X | | | X | 54 |
| ID 3886 | Dharmshale et al. 2015 | India | 23 | 23 | Not mentioned | Single-center, hospital-based study | clinical, microscopic examination, culture | | X | | | | | | X | | X | X | 60 |

*(Continued)*

Table 1. (Continued)

| ID | Authors | Country | Total Mycetoma patients | Total study population | Study design | Study Population Source | Diagnosis tools | Age | Gender | Education | Residency | Occupation | Comorbidities | Severe disease | Body region | Outcomes | Treatment | Causative Organism | Citation |
|---|---|---|---|---|---|---|---|---|---|---|---|---|---|---|---|---|---|---|---|
| ID 1887 | El Hag et al. 1994 | Sudan | 14 | 14 | Not mentioned | Single-center, hospital-based study | clinical, microscopic examination | X | X | | | | | | | X | | | X | 38 |
| ID 4072 | El Shamy et al. 2012 | Sudan | 42 | 42 | Prospective observational | Single-center, hospital-based study | clinical, imaging, microscopic examination, cytology, culture | X | X | | | | | X | X | | | X | 64 |
| ID 5776 | Elgallali et al. 2010 | Tunisia | 15 | 15 | Retrospective observational | Single-center, hospital-based study | clinical, microscopic examination, culture | X | X | | X | | | | X | X | X | X | 81 |
| ID 5772 | Enbiale et al. 2023 | Ethiopia | 118 | 143 | Retrospective observational | Multi-center, hospital-based study | clinical, microscopic examination, culture | | | | | | | | | | | X | 79 |
| ID 963 | Estrada-Castañón et al. 2019 | Mexico | 113 | 113 | Retrospective observational | Single-center, hospital-based and community-based study | clinical, imaging, culture, microscopic examination | X | X | | | X | | | X | | X | X | 29 |
| ID 445 | Fahal et al. 2014 | Sudan | 33 | 33 | Prospective observational | Community-based study | histopathology, culture | X | X | | X | X | | | X | X | X | X | 21 |
| ID 5253 | Fahal et al. 2015 | Sudan | 6792 | 6792 | Cross-sectional | Single-center, hospital-based study | clinical, imaging, serology, culture, microscopic examination | X | X | | | X | X | | X | X | X | X | 71 |
| ID 3805 | Ganawa et al. 2021 | Sudan | 594 | 594 | Retrospective observational | Community-based study | clinical, microscopic examination, culture | | X | | | X | | | | | | X | 17 |
| ID 2396 | Guimarães et al. 2003 | Brazil | 40 | 40 | Retrospective observational | Single-center, hospital-based study | culture, microscopic examination | | X | | | | | | X | | | X | 42 |
| ID 3423 | Hashemi et al. 2005 | Iran | 62 | 62 | Not mentioned | Single-center, hospital-based study | Not mentioned | | X | | X | X | | | X | | | X | 51 |
| ID 2946 | Hassan et al. 2021 | Sudan | 6983 | 6983 | Retrospective observational | Single-center, hospital-based study | full coverage | X | X | X | X | X | | | | | | X | 47 |
| ID 2183 | Hassan et al. 2022 | Sudan | 359 | 1436 | Case-control | Multi-center, hospital-based and community-based study | clinical, microscopic examination | X | X | | | | | | X | | | | 40 |

*(Continued)*

**Table 1.** (Continued)

| ID | Authors | Country | Total Mycetoma patients | Total study population | Study design | Study Population Source | Diagnosis tools | Age | Gender | Education | Residency | Occupation | Comorbidities | Severe disease | Body region | Outcomes | Treatment | Causative Organism | Citation |
|---|---|---|---|---|---|---|---|---|---|---|---|---|---|---|---|---|---|---|---|
| ID 5654 | Hay et al. 1982 | United Kingdom | 43 | 41 | Not mentioned | Single-center, hospital-based study | clinical, culture, serology, microscopic examination | | X | | | | | | X | | | X | 76 |
| ID 5783 | Kaliswaran et al. 2003 | India | 25 | 25 | Prospective observational | Single-center, hospital-based study | clinical, microscopic examination, culture, imaging | | X | | | | | | X | X | X | X | 83 |
| ID 5516 | Kallel et al. 2004 | Tunisia | 13 | 13 | Cross-sectional | Single-center, hospital-based study | clinical, imaging, culture, microscopic examination | | X | | X | X | | X | X | | X | X | 74 |
| ID 509 | Kamalam et al. 1976 | India | 5 | 4103 | Prospective observational | Single-center, hospital-based study | clinical, microscopic examination, culture | X | X | | | | | | X | | | X | 24 |
| ID 510 | Kébéa et al. 2021 | Mauritania | 87 | 87 | Retrospective observational | Multi-center, hospital-based study | clinical, imaging, culture, microscopic examination | | X | | | X | | | X | | X | X | 25 |
| ID 951 | Khatri et al. 2002 | Yemen | 70 | 70 | Prospective observational | Single-center, hospital-based study | clinical, imaging, microscopic examination | | X | | | | | | | X | X | X | 28 |
| ID 3443 | Khatri et al. 2021 | Yemen | 184 | 184 | Prospective observational | Single-center, hospital-based study | clinical, imaging, culture, microscopic examination | | X | | | | | X | X | X | X | X | 52 |
| ID 3501 | Kizera et al. 2020 | Uganda | 249 | 249 | Cross-sectional | Single-center, hospital-based study | clinical, microscopic examination | | X | | | | | X | X | X | X | X | 56 |
| ID 5770 | Kunna et al. 2020 | Sudan | 389 | 389 | Cross-sectional | Single-center, hospital-based study | Not mentioned | | X | X | X | X | X | | | X | X | | 77 |
| ID 5796 | Lewall et al. 1985 | Saudi Arabia | 30 | 30 | Retrospective observational | Single-center, hospital-based study | Not mentioned | | X | | | | | | | | | | 86 |
| ID 1641 | Lopez et al. 2023 | Mexico | 70 | 70 | Retrospective observational | Single-center, hospital-based study | clinical, culture, microscopic examination | | X | | | X | | | X | | | X | 34 |
| ID 5784 | Lopez-Martinez et al. 2013 | Mexico | 3933 | 3933 | Not mentioned | Multi-center, hospital-based study | Not mentioned | | X | | | X | | | X | | | X | 84 |

(Continued)

Table 1. (Continued)

| ID | Authors | Country | Total Mycetoma patients | Total study population | Study design | Study Population Source | Diagnosis tools | Age | Gender | Education | Residency | Occupation | Comorbidities | Severe disease | Body region | Outcomes | Treatment | Causative Organism | Citation |
|---|---|---|---|---|---|---|---|---|---|---|---|---|---|---|---|---|---|---|---|
| ID 1648 | Maiti et al. 2002 | India | 264 | 264 | Retrospective observational | Single-center, hospital-based study | clinical, culture, microscopic examination | | X | | | X | | | X | | | X | 35 |
| ID 4290 | Mallick et al. 2021 | Brazil | 12 | 12 | Retrospective observational | Single-center, hospital-based study | clinical, microscopic examination, culture | X | X | | X | X | | | X | X | X | X | 66 |
| ID 1640 | Maritinez et al. 1992 | Mexico | 2105 | 2105 | Retrospective observational | Multi-center, hospital-based study | Not mentioned | | X | | | X | | | X | | | X | 33 |
| ID 4424 | Méndez-Tovar et al. 2021 | Mexico | 36 | 36 | Cross-sectional | Single-center, hospital-based study | clinical, microscopic examination, culture | | X | | | X | X | | X | | X | X | 68 |
| ID 3140 | Mufti et al. 2015 | Saudi Arabia | 19 | 19 | Retrospective observational | Single-center, hospital-based study | clinical, culture, microscopic examination | X | X | | | | | X | X | X | X | X | 49 |
| ID 3554 | Musa et al. 2022 | Sudan | 503 | 503 | Cross-sectional | Single-center, hospital-based study | clinical, culture, microscopic examination | | X | | | X | | | X | X | X | X | 57 |
| ID 137 | Ndiaye et al. 2011 | Senegal | 113 | 113 | Retrospective observational | Single-center, hospital-based study | clinical, microscopic examination, culture | | X | | | X | | | X | | | X | 19 |
| ID 2922 | Negroni et al. 1998 | Argentina | 54 | 54 | Retrospective observational | Single-center, hospital-based study | histology, microscopic examination | | X | | | | | | | X | X | X | 46 |
| ID 1689 | Negroni et al. 2006 | Argentina | 76 | 76 | Not mentioned | Single-center, hospital-based study | clinical, culture, microscopic examination | | X | | | | | | X | X | X | X | 36 |
| ID 3458 | Padhi et al. 2010 | India | 13 | 13 | Retrospective observational | Single-center, hospital-based study | clinical, imaging, culture, microscopic examination | X | X | | | X | | X | X | X | X | X | 53 |
| ID 4716 | Sampaio et al. 2017 | Brazil | 21 | 21 | Retrospective observational | Single-center, hospital-based study | clinical, imaging, microscopic examination | | X | | | | X | X | X | X | X | X | 69 |
| ID 5780 | Sarr et al. 2015 | Senegal | 44 | 44 | Retrospective observational | Single-center, hospital-based study | Not mentioned | | X | | | | | | | X | | | 82 |

(Continued)

**Table 1.** (Continued)

| ID | Authors | Country | Total Mycetoma patients | Total study population | Study design | Study Population Source | Diagnosis tools | Age | Gender | Education | Residency | Occupation | Comorbidities | Severe disease | Body region | Outcomes | Treatment | Causative Organism | Citation |
|---|---|---|---|---|---|---|---|---|---|---|---|---|---|---|---|---|---|---|---|
| ID 3864 | Sawatkar et al. 2019 | India | 11 | 11 | Retrospective observational | Clinical-based | clinical, imaging, microscopic examination, culture | X | X | | X | X | | X | X | | X | X | 59 |
| ID 2780 | Sear et al. 2022 | Venezuela | 16 | 16 | Retrospective observational | Single-center, hospital-based study | clinical, culture, microscopic examination | X | X | | | X | | | X | | | X | 44 |
| ID 3494 | Siblany et al. 1998 | Saudi Arabia | 36 | 36 | Retrospective observational | Multi-center, hospital-based study | clinical imaging, microscopic examination | | X | | | X | | X | X | X | X | X | 55 |
| ID 5230 | Siddig et al. 2016 | Sudan | 50 | 50 | Cross-sectional | Single-center, hospital-based study | clinical, microscopic examination, cytology | X | X | | | | | | | | | X | 70 |
| ID 5411 | Siddig et al. 2022 | Sudan | 72 | 80 | Cross-sectional | Single-center, hospital-based study | clinical, imaging, genetic culture, microscopic examination | | | | | | | | | | | X | 73 |
| ID 3340 | Sow et al. 2020 | Senegal | 193 | 193 | Retrospective observational | Multi-center, hospital-based study | clinical, imaging, culture, microscopic examination | X | X | | | X | | X | X | X | X | X | 50 |
| ID 1263 | Sran et al. 1973 | India | 110 | 110 | Retrospective observational | Single-center, hospital-based study | microscopic examination | | | | | X | | X | | X | X | | 32 |
| ID 2678 | Traore et al. 2021 | Mali | 19 | 19 | Retrospective observational | Single-center, hospital-based study | clinical, imaging, microscopic examination | | X | | | X | | X | X | X | X | X | 43 |
| ID 4423 | Venugopal et al. 1975 | India | 90 | 90 | Not mentioned | Single-center, hospital-based study | clinical, microscopic examination, culture | | X | | | | | | X | | | X | 67 |
| ID 1709 | Venugopal et al. 1990 | Saudi Arabia | 23 | 23 | Retrospective observational | Multi-center, hospital-based study | clinical, culture, microscopic examination | | X | | | | | | X | | | X | 37 |
| ID 456 | Yousif et al. 2009 | Sudan | 230 | 240 | Prospective observational | Single-center, hospital-based study | microscopic examination | X | X | | | | | | X | | | X | 22 |
| ID 223 | Zein et al. 2012 | Sudan | 1544 | 1544 | Prospective observational | Single-center, hospital-based study | clinical, microscopic examination, culture, imaging | | X | | | | | | | X | | | 20 |

the 72 studies comprised 22 countries: Sudan (n = 20), India (n = 10), Mexico (n = 7), Brazil (n = 5), Saudi Arabia (n = 4), Tunisia (n = 4), Senegal (n = 3) Argentina (n = 2), Venezuela (n = 2) and Yemen (n = 2). One study was conducted in each of the following countries: Bulgaria, Côte d'Ivoire, Ethiopia, Iran, Kenya, Mali, Mauritania, Morocco, Togo, Uganda, United Arab Emirates, and the United Kingdom. Additionally, one study was conducted in West Africa across many countries, but it did not report the number of mycetoma patients per country. The number of cases reported from each country ranged from 12 (Morocco) to 16,345 cases (Sudan; S3 Table). Most studies were retrospective observational (n = 34/72). The studies varied in the source of the population used for the study, including patients from a hospital system (n = 67/72) and community-wide surveys (n = 5/72). The most common diagnostic tools included clinical observation, microscopic examination, culture, and/or imaging, while cytology, histopathology, serology, or genetic tools were less common.

## Sociodemographic and clinical characteristics

Key patterns were identified from analysing the sociodemographic and clinical characteristics of mycetoma patients across 69 studies (Table 2). The mean age reported across the studies was 35.6 years (range: 27.0–53.2 yrs, N = 20 studies) (Fig 3A). Patients with mycetoma were predominantly male (mean percent: 73.5%, range: 40.0-100.0%, N = 68) (Fig 3B). Few studies reported on patients' education level (N = 3 studies) and residence characteristics (N = 9). More patients were literate (66.7%, 61.3–69.7%) than illiterate (33.1%, 29.8–38.7%) and resided in rural settings (72.8%, 39.3–100.0%) than urban settings (16.6%, 0.0–56.0%), however, many patients' education level and residence status were not reported. Farming was the most common occupation, but occupations varied widely among the included studies (38.3%, 6.3–72.2%, N = 68, Fig 3C). Most other occupations represented between 14.0% and 22.2% of patients, while professional and government occupations tended to represent relatively lower mean percents (3.3% and 5.2%, respectively). Thirty-nine and three-tenths of patients had severe disease with bone and muscle involvement (range: 2.2–76.9%, N = 19), reflecting the disparity among the studies. Most patients had disease of the lower limbs (76.9%, 9.3–100.0%, N = 57; Fig 4A). A lower mean percent of patients had clinical manifestations of mycetoma in other body regions (5.3–12.8%); in some studies, a high percentage of patients reported disease in the trunk, buttock, and groin regions.

The outcomes were not reported for many patients due to loss to follow-up or other reasons (N = 28 studies). Few studies reported on patient remission or improvement status. Among those that did report patient outcomes, a mean of 33.4% of patients were cured (2.2–97.0%), 14.7% had amputations (0.8–63.2%), and 17.6% had recurrence (0.0–52.8%, Fig 4B). While many patients received multiple forms of treatment, most received pharmaceutical treatment (92.1%, 36.8–100.0%, N = 30). Forty and six tenths percent had surgical excision (2.3–97.0%). The mean percent of patients reported using religious or traditional herbalist treatments was 51.2% and 68.1%, respectively, however, only a few studies reported on those treatment interventions. Among the five studies that reported on comorbidities, 14.5% of patients had hypertension (0.3–33.3%) and 14.5% had diabetes (0.5–38.8%), but there was high variability among studies and low sample size (N = 5).

## Causative organisms' distribution

Most cases identified in Africa and the Arabian Peninsula, including Sudan, Uganda, Togo, Mali, Senegal, Saudi Arabia, Côte d'Ivoire, and Yemen, were classified as eumycetoma (Fig 5 and Table 3). Mexico, Iran, and India have more cases classified as actinomycetoma. Many other countries had a relatively even mix of both aetiologies, including Brazil, Argentina, and Tunisia. Other countries had a large proportion of patients where the causative organism was not identified (unspecified) or not reported.

The most common bacterial species were *Actinomadura madurae* (N = 1180 patients), *Actinomadura pelletieri* (N = 204), *Nocardia brasiliensis* (N = 4365), *Nocardia asteroides* (N = 120), *Streptomyces somaliensis* (N = 1198), and the most common fungal species were *Madurella mycetomatis* (N = 5924) and *Trematosphaeria grisea* (N = 143) (S4 Table). The diversity of causative organism genera identified differs across countries (Fig 6). Notably, Mexico (n = 15 genera), India (n = 12),

**PLOS** Neglected Tropical Diseases

**Table 2.** Mean percent and range of mycetoma patients reported from the reviewed studies for each sociodemographic and clinical characteristic. Where fewer than the total number of studies reported a specific category, we include the number of studies used to calculate the mean percentage in parentheses.

| Number of studies | Mean percent | Range |
|---|---|---|
| **Age** (n = 8,238 patients; N = 20 studies | | |
| mean yrs | 35.61 | (27.0, 53.2) |
| **Sex** (n = 22,045; N = 68) | | |
| Male | 73.5 | (40.0, 100.0) |
| Female | 26.2 | (0.0, 60.0) |
| Not reported | 0.4 | (0.0, 11.1) |
| **Education** (n = 774; N = 3) | | |
| Illiterate | 33.1 | (29.8, 38.7) |
| Literate | 66.7 | (61.3, 69.7) |
| **Residency** (n = 871; N = 9) | | |
| Urban | 16.6 | (0.0, 56.0) |
| Rural | 72.8 | (39.3, 100.0) |
| Not reported | 10.6 | (0.0, 46.7) |
| **Occupation** (n = 17,722; N = 36) | | |
| Animal Breeder, (7 studies reported) | 19.1 | (1.4, 42.5) |
| Farmer (35) | 38.3 | (6.3, 72.2) |
| Freelancer (13) | 11.4 | (0.9, 49.4) |
| Government (11) | 5.2 | (0.9, 15.7) |
| Housewife (22) | 15.2 | (5.3, 31.8) |
| NonSkilful Worker (6) | 15.7 | (1.0, 39.0) |
| Other Jobs, Not specified (21) | 17.7 | (0.7, 54.8) |
| Professional (3) | 3.3 | (0.3, 5.3) |
| Skilful Worker (21) | 14 | (2.2, 33.3) |
| Student (22) | 15.3 | (1.8, 33.8) |
| Unemployed (10) | 8.1 | (0.9, 29.3) |
| Not reported (18) | 22.2 | (0.4, 84.8) |
| **Comorbidities** (n = 7,412; N = 5) | | |
| Hypertension (4) | 14.5 | (0.3, 33.3) |
| Diabetes (4) | 14.5 | (0.5, 38.8) |
| Renal Disease (1) | 0.1 | (0.1, 0.1) |
| HIV (1) | 4.8 | (4.8, 4.8) |
| TB (1) | 0.1 | (0.1, 0.1) |
| Asthma (1) | 4.8 | (4.8, 4.8) |
| Leprosy (1) | 0.04 | (0.04, 0.04) |
| Thyroid (1) | 1.1 | (1.1, 1.1) |
| **Severe disease** | | |
| (n = 1,208; N = 19) | 39.3 | (2.2, 76.9) |
| **Involved body regions** (n = 17,594; N = 57) | | |
| Buttocks & Groin (22) | 8.8 | (0.4, 73.6) |
| Face & Neck (30) | 5.3 | (0.4, 40.0) |
| Lower Limbs (56) | 76.9 | (9.3, 100.0) |
| Multiple sites (5) | 2.4 | (1.80, 3.8) |
| Trunk (32) | 12.8 | (0.7, 68.2) |

*(Continued)*

**Table 2.** (Continued)

| Number of studies | Mean percent | Range |
|---|---|---|
| Upper Limbs (49) | 9.4 | (1.8, 22.4) |
| Unspecified (19) | 10.7 | (0.3, 31.8) |
| **Disease Outcomes** (n = 10,912; N = 28) | | |
| Cured (14) | 33.4 | (2.2, 97.0) |
| Remission (2) | 42.8 | (41.7, 44.0) |
| Improved (2) | 62.5 | (59.3, 65.7) |
| Amputation (23) | 14.7 | (0.8, 63.2) |
| Recurrence (11) | 17.6 | (0.0, 52.8) |
| Not Improved (3) | 18.7 | (7.4, 33.3) |
| Lost to follow-up (14) | 22.3 | (0.0, 54.3) |
| Not reported (28) | 43.6 | (0.0, 99.2) |
| **Treatment Received** (n = 9,058; N = 30) | | |
| Pharmaceutical (28) | 92.1 | (36.8, 100.0) |
| Surgical debridement (1) | 76.9 | (76.9, 76.9) |
| Surgical excision (16) | 40.6 | (2.3, 97.0) |
| Surgical amputation (19) | 14 | (0.8, 63.2) |
| Surgical resection (2) | 27 | (25.9, 28.0) |
| Religious (1) | 51.2 | (51.2, 51.2) |
| Herbal traditional (3) | 68.1 | (52.6, 80.8) |

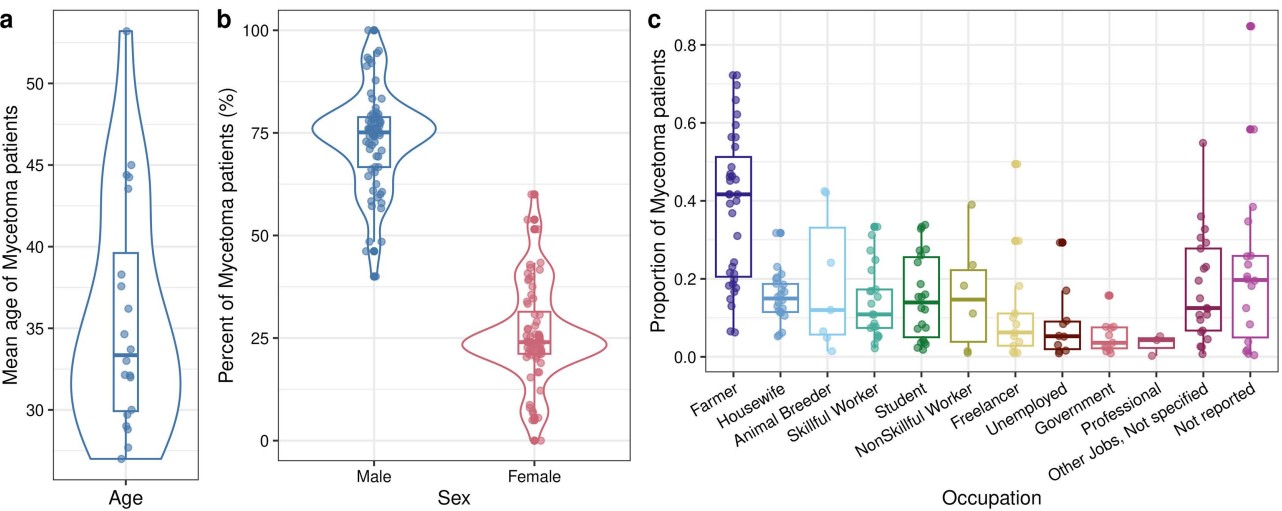

**Fig 3. Sociodemographic characteristics of mycetoma patients from the reviewed studies.** (a) Mean age of reported mycetoma patients, (b) percent of males and females affected by mycetoma reported, and (c) percent of mycetoma patients with reported occupations. Each point represents a study-reported value for the given characteristic; not all studies reported the same sets of characteristics, therefore, the number of points differs among different occupations (N studies reported in Table 2). Colours correspond to the x-axis labels in each panel to allow for easier differentiation of categories. Boxplots show the median and interquartile ranges of the points, and the violin plots (panels a and b) show the relative density of points across the full range, where greater width indicates higher density.

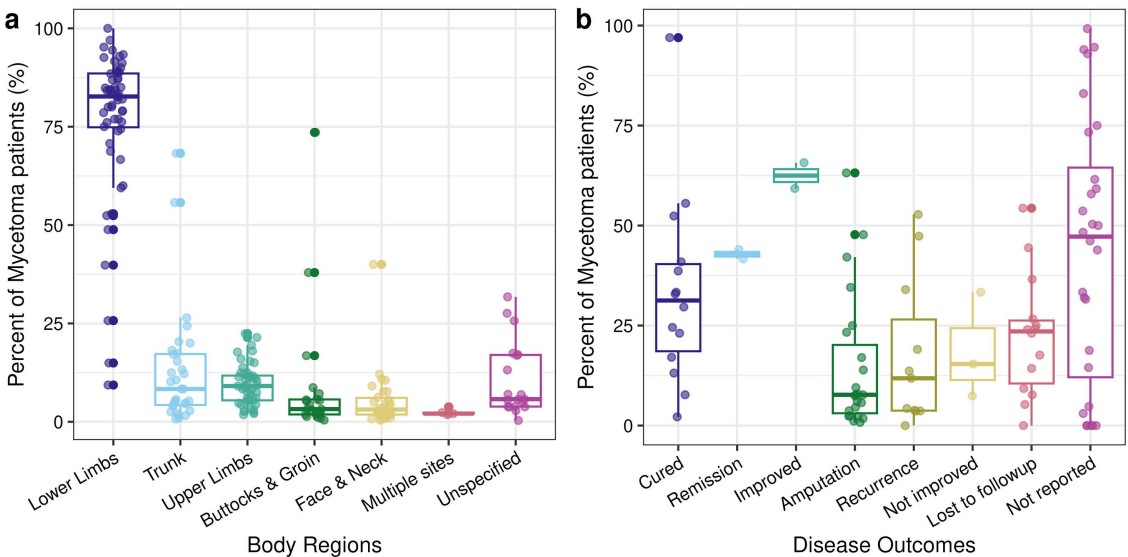

**Fig 4. Clinical characteristics of mycetoma patients from the reviewed studies. (a)** The percentage of mycetoma patients with reported involvement of each body region, and **(b)** the percentage of mycetoma patient outcomes reported. Each point represents a study-reported value for the given characteristic; not all studies reported the same sets of characteristics, therefore, the number of points differs among different body regions and outcomes (N studies reported in Table 2). Colours correspond to the x-axis labels in each panel to allow for easier differentiation of categories. Boxplots show the median and interquartile ranges of the points.

Proportion of Mycetoma patients reported by aetiology per country

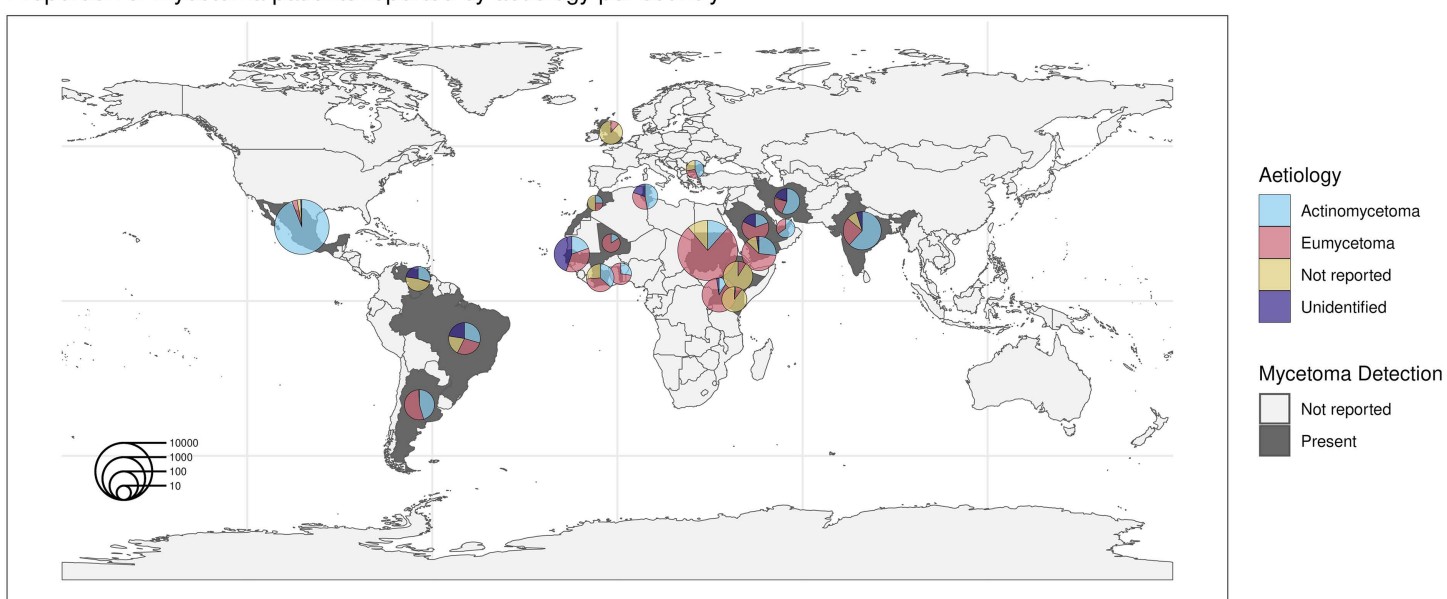

**Fig 5. Proportion of reported mycetoma cases where the causative organism was identified as actinomycetoma or eumycetoma per country.** Pie charts are sized by the total number of cases detected within the country. Countries are dark grey if a selected study reported mycetoma patients from that country, or light grey if none of the selected studies reported mycetoma patients from that country. Basemap source: Made in R with Natural Earth. Free vector and raster map data @ http://naturalearthdata.com/.

**Table 3. Counts of actinomycetoma, eumycetoma, not reported, and unidentified cases per country from the reviewed studies.**

| Country | Actinomycetoma | Eumycetoma | Not reported | Unidentified |
|---|---|---|---|---|
| Argentina | 59 | 70 | 1 | 0 |
| Brazil | 38 | 36 | 26 | 29 |
| Bulgaria | 8 | 5 | 5 | 0 |
| Cote d'Ivoire | 32 | 32 | 23 | 0 |
| Ethiopia | 0 | 10 | 102 | 0 |
| India | 290 | 114 | 42 | 23 |
| Iran | 35 | 14 | 1 | 12 |
| Kenya | 1 | 5 | 52 | 0 |
| Mali | 3 | 16 | 0 | 0 |
| Mexico | 6088 | 201 | 138 | 45 |
| Morocco | 3 | 3 | 6 | 0 |
| Saudi Arabia | 14 | 45 | 0 | 13 |
| Senegal | 60 | 109 | 0 | 137 |
| Sudan | 1725 | 11476 | 1662 | 8 |
| Togo | 9 | 24 | 0 | 0 |
| Tunisia | 24 | 17 | 0 | 10 |
| Uganda | 22 | 221 | 0 | 6 |
| United Arab Emirates | 13 | 7 | 0 | 0 |
| United Kingdom | 0 | 5 | 38 | 0 |
| Venezuela | 15 | 1 | 25 | 12 |
| West Africa | 12 | 15 | 0 | 19 |
| Yemen | 68 | 154 | 26 | 6 |

Proportion of Mycetoma patients identified to genus per country

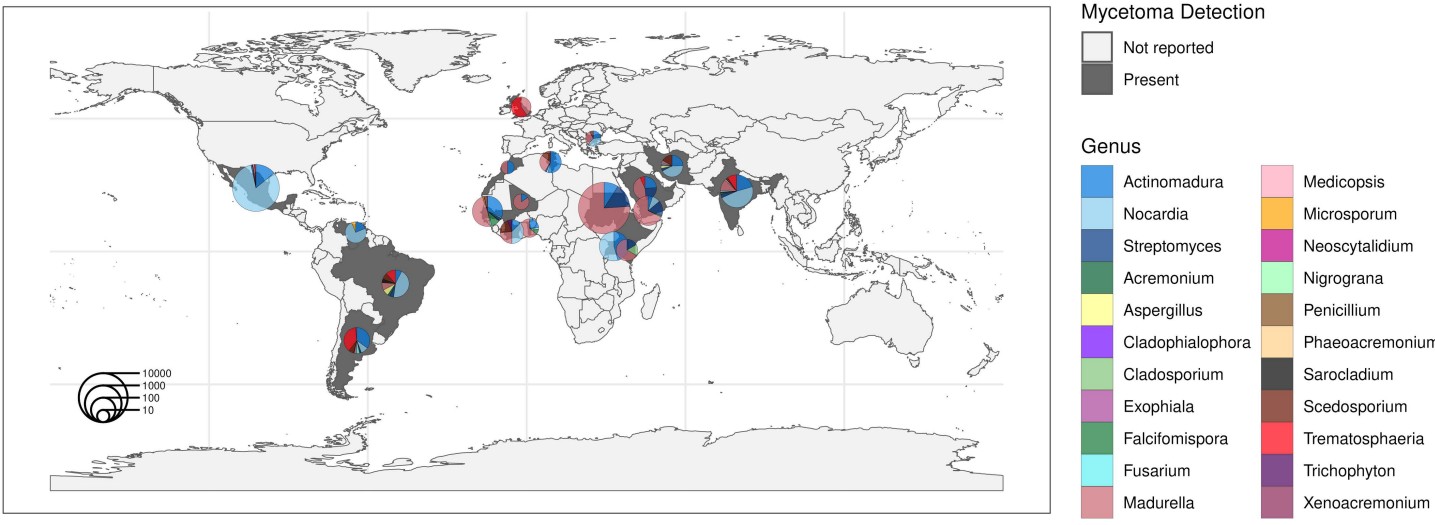

**Fig 6. Proportion of reported mycetoma cases where the causative organism was identified to the genus for each country.** Pie charts are sized by the total number of cases detected within the country. The pieces of the pie charts exclude patients where the causative organism was not reported or not identified to the genus level. Countries are dark grey if a selected study reported mycetoma patients from that country, or light grey if none of the selected studies included that country. Ethiopia and the United Arab Emirates did not have any isolates identified beyond the actinomycetoma and eumycetoma aetiologies, and therefore, they do not have pie charts shown here. Basemap source: Made in R with Natural Earth. Free vector and raster map data @ http://naturalearthdata.com/.

Senegal (n = 11), Argentina (n = 10), Brazil (n = 9), and Iran (n = 8) had over eight genera reported, though some genera had relatively few cases. Most other countries in this study reported between two and five genera of mycetoma-causative organisms.

## Discussion

We found that the typical profile of patients with mycetoma was generally adult men and predominantly farmers. This profile aligns with previous literature [5,8,86,87], as male predominance is attributed to outdoor activities and potential genetic and hormonal factors [5,88,89]. Interestingly, one community-based study reported a slightly higher prevalence among women [86], where in this case, female care-seeking behaviour may explain the discrepancy since women often delay seeking medical care and present at advanced stages. In the present study, the most affected age group is 20–50, which is the typical age group for mycetoma based on prior studies [5,8,86,87], as this age group is the most active and engaged in agricultural activities. The occupational exposure of farmers highlights the role of agriculture as a key risk factor, especially in rural areas where there is heightened exposure to the causative organisms [87,90–92].

The clinical profile of patients with mycetoma reflects previous findings across the literature, identifying the lower limbs as a primary involved body region [87,93]. Lower limbs are most likely to be exposed to soil-borne pathogens, especially in regions where people are usually bare-footed during their activities [15,30]. These findings suggest that an important preventive measure to reduce mycetoma risk is lower-extremity protection from injuries and soil exposure in rural, endemic regions [19].

We found that most mycetoma patients had received pharmaceutical treatment. Surgical interventions were relatively less common than pharmaceutical treatment, as this regimen is more common for actinomycetoma, but surgical and pharmaceutical treatments were often used in combination. This aligns with treatment modalities commonly used for eumycetoma [94]. Furthermore, the variability in the disease outcomes that we observed highlights the complexity of disease treatment. For example, the high rate of amputations indicates the severe disease morbidity associated with either untreated, delayed, or inadequately managed mycetoma lesions [18]. Meanwhile, recurrence rates are likely influenced by the inadequate follow-up, poor accessibility or adherence to medications, and underlying comorbidities [18]. Low reporting of comorbidities in the literature limits our understanding of how comorbidities influence the progression of the disease. For instance, delayed wound healing in diabetic mycetoma patients could complicate the treatment and increase the severity of the lesion.

One of the main challenges identified was the inconsistency in publications including sociodemographic and clinical data about patients with mycetoma, which complicated our ability to establish a clear profile of mycetoma patients. For example, educational status was only mentioned in three studies, comorbidities were reported in five studies, and residence was reported in nine studies. Furthermore, less than half of the included studies reported key patient characteristics such as age, severity of disease, disease outcomes, and treatment received. Insufficient reporting of these characteristics of mycetoma patients suggests a lack of accurate profiling for the disease and makes comparability across studies particularly challenging. It is crucial to standardise reporting practices for mycetoma patients' characteristics in research studies to ensure consistency and comparability among studies. Consistent reporting of mycetoma patient characteristics can improve our understanding of the mycetoma patient profile to further enhance identifying patients at risk for mycetoma exposure and the diagnosis of infected patients. We recommend that global health bodies or relevant stakeholders develop standardised guidelines for mycetoma surveillance. Key patient characteristics to consider reporting in future studies include the demographic profile (sex, education, residence, occupation), clinical characteristics (comorbidities, disease severity, body site(s) involved, outcome, treatment plan used), and the causative organism (bacteria or fungus identified to the genus and species level).

Mycetoma was found to have a global spread focused in tropical and subtropical regions known as the mycetoma belt. Due to heterogeneity of study populations and variations in reporting across studies, calculating the global prevalence

proved extremely difficult. Nevertheless, we found that there are distinct patterns in the geographic distribution of the causative organisms. Regional variability in genus distribution shows that eumycetoma (fungal) was primarily reported in Africa and the Arabian Peninsula, while actinomycetoma (bacterial) was the predominant causative organism in Mexico, Iran and India. The most commonly reported eumycetoma genera were *Madurella*, *Trematosphaeria*, and *Scedosporium*. Eighteen genera and 32 species were identified among the studies, but 6,327 patients had a fungal mycetoma that was not identified to the genus or species. Eumycetoma were frequently reported in Sudan, Senegal and Somalia [11,15,17,18,48]. Several environmental predictors may contribute to the risk of fungal mycetoma, such as the presence of thorny trees [90–92]. In Sudan, for example, it appears that the ecological niche of *Acacia* trees overlaps with mycetoma-causative organisms and may be implicated in exposure to mycetoma through *Acacia* thorns, since *Madurella mycetomatis* has been found to be present on *Acacia* thorns [10,90,91]. Some etiological agents were reported under historical names, for example, *Nocardia asteroides* is now a species complex including several species, *Madurella pseudomycetomatis* is more common than *M. mycetomatis* in the Americas, and *Trematosphaeria grisea* may correspond to *Nigrograna mackinonii*, and *Trichophyton Soudanense* is now under the name *Trichophyton rubrum*.

Actinomycetoma had only three genera (11 species) identified, including *Actinomadura*, *Nocardia*, and *Streptomyces*, among the evaluated studies, with most cases identified as *Actinomadura madurae, Nocardia brasiliensis,* and *Streptomyces somaliensis* in Mexico, Brazil, and other parts of South America. In contrast to eumycetoma, most actinomycetoma was at least identified to the genus-level for most cases, with only 13 cases reported as unidentified actinomycetoma [27,31,32,40,43,46]. The distribution of mycetoma appears to be influenced by environmental factors. Eumycetoma is strongly associated with thorny trees, particularly acacia, and proximity to rivers. Actinomycetoma, on the other hand, is more affected by the distance to water bodies like lakes and ponds and tends to be sensitive to colder temperatures [92].

Overall, there were 2,147 and 320 cases of unreported and unidentified causative organisms, respectively. Overcoming barriers in laboratory capacity to allow for greater identification and reporting of mycetoma causative organisms could improve our understanding of the geographic distribution, ecological niches, and exposure pathways for each causative organism species.

The study has several limitations. First, the heterogeneity in the population sources and inconsistent data reporting among studies limited our ability to compare disease epidemiology over time and between regions, which makes it difficult to draw conclusions about potential risk factors. Second, there is a possibility for case duplication in studies from overlapping regions and similar time frames, however, we removed studies that had known case duplication to minimise this issue. Third, studies with lower quality scores were included to prevent data loss and ensure a comprehensive dataset on a rare disease. While this may have influenced the overall findings by increasing the risk of bias, generally low-scoring studies were primarily missing methodological details rather than lacking validity characteristics. Additionally, the publication of these studies with lower quality scores in recognised peer-reviewed journals lends them credibility, suggesting that the findings still add value to understand the disease patterns. Finally, the limited number of publications from a relatively small number of countries likely does not represent the underlying population globally.

Our findings inform targeted public health interventions by identifying high-risk populations and endemic regions. The demonstrated gap in disease reporting reflects the need for standardised data reporting and strong surveillance systems. Health authorities should implement preventive measures for at-risk groups in endemic areas and promote awareness campaigns in vulnerable communities. Understanding the disease prevalence, patients' characteristics, and healthcare resources gaps is essential for designing effective prevention and care strategies.

In conclusion, this systematic review is the first to profile the sociodemographic and clinical characteristics of patients with mycetoma while mapping the geographic distribution of causative organisms at the genus level. Unlike prior studies, our study utilised an extensive search strategy with no restrictions on language or timing of publication. This approach minimises bias and enables the inclusion of 72 eligible studies, providing a solid foundation for future research. A better understanding of mycetoma epidemiology will enable more targeted education, prevention, and treatment in high-risk

areas, ultimately reducing the disease burden. Our findings highlight the need for standardised data collection, consistent reporting, and regular calculation of prevalence across future studies.

While this review included studies of varying quality to ensure comprehensive data, we encourage future researchers to prioritise methodological transparency, which was the primary reason for low scores in the Newcastle-Ottawa Scale, particularly in comparability and selection metrics. Additionally, we found that many fungal mycetoma cases lacked genus or species-level identification, highlighting the need to improve fungal diagnostic capacity and access. Mapping the distribution of actinomycetoma, eumycetoma, and their respective causative organisms can enhance clinicians' awareness of the most likely pathogens in their regions, leading to more accurate diagnoses and appropriate treatments.

## Supporting information

**S1 Table. The full search strategy with all the included search terms.**
(XLSX)

**S2 Table. Newcastle-Ottawa quality scores for all included studies.**
(XLSX)

**S3 Table. Total number of mycetoma patients reported per country from 25,513 patients included in 58 studies where the causative organism was identified.**
(XLSX)

**S4 Table. The total number of mycetoma patients reported for each causative organism species and genus was 25,513 patients, and they were included in 58 studies where the causative organism was identified.**
(XLSX)

**S5 Table. Basic characteristic table.**
(XLSX)

**S1 Fig. Quality assessment traffic light.**
(PNG)

**S2 Fig. PRISMA checklist 2020.**
(PDF)

## Acknowledgments

We would like to thank Shawheen J. Rezaei for his insightful comments and edits on the manuscript. We would also like to thank Dr. Mojahid Mohamed Alhussein for his assistance in acquiring access to some of the included studies' full texts and his insightful additions to the paper.

The findings and conclusions in this report are those of the authors and do not necessarily represent the official position of the Centers for Disease Control and Prevention.

## Author contributions

**Conceptualization:** Mohamed Elmuntasir Salah, Kirlus Habib, Fadila Alhamwi, Suad Abdelwahab, Ahmed Fahal.

**Data curation:** Mohamed Elmuntasir Salah, Kirlus Habib, Yassin Ahmed.

**Formal analysis:** Mohamed Elmuntasir Salah, Michelle L. Fearon Scales, Dallas J. Smith.

**Investigation:** Mohamed Elmuntasir Salah, Kirlus Habib, Yassin Ahmed.

**Methodology:** Mohamed Elmuntasir Salah, Michelle L. Fearon Scales, Kirlus Habib, Fadila Alhamwi, Suad Abdelwahab, Yassin Ahmed, Dallas J. Smith.

**Project administration:** Mohamed Elmuntasir Salah, Michelle L. Fearon Scales, Dallas J. Smith, Ahmed Fahal.

**Resources:** Mohamed Elmuntasir Salah, Michelle L. Fearon Scales, Dallas J. Smith, Ahmed Fahal.

**Software:** Michelle L. Fearon Scales.

**Supervision:** Mohamed Elmuntasir Salah, Michelle L. Fearon Scales, Kirlus Habib, Dallas J. Smith, Manal Mohamed Khalid, Ahmed Fahal.

**Validation:** Mohamed Elmuntasir Salah, Michelle L. Fearon Scales, Kirlus Habib, Fadila Alhamwi, Suad Abdelwahab, Manal Mohamed Khalid.

**Visualization:** Mohamed Elmuntasir Salah, Michelle L. Fearon Scales, Dallas J. Smith.

**Writing – original draft:** Michelle L. Fearon Scales, Fadila Alhamwi, Suad Abdelwahab.

**Writing – review & editing:** Mohamed Elmuntasir Salah, Michelle L. Fearon Scales, Kirlus Habib, Fadila Alhamwi, Suad Abdelwahab, Dallas J. Smith, Manal Mohamed Khalid, Ahmed Fahal.

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
