## [Decision Letter · Decision Letter 0]

30 Apr 2025

Response to Reviewers
Revised Manuscript with Track Changes
Manuscript

Shaden Kamhawi

co-Editor-in-Chief

Paul Brindley

co-Editor-in-Chief

**Additional Editor Comments:**

Methods section. I suggest including a brief explanation of the heterogeneity of the studies.

Results section. In Table S4 they include the cases of actinomycetoma and eumycetoma. This information should be highlighted in the manuscript. Therefore, using the same variables as in Table 2, I suggest including a Table (Table 3) describing actinomycetoma vs. eumycetoma cases.

**Journal Requirements:**

1) Please upload all main figures as separate Figure files in .tif or .eps format. For more information about how to convert and format your figure files please see our guidelines: 

2) We have noticed that you have uploaded Supporting Information files, but you have not included a list of legends. Please add a full list of legends for your Supporting Information files after the references list.

3) Some material included in your submission may be copyrighted. According to PLOSu2019s copyright policy, authors who use figures or other material (e.g., graphics, clipart, maps) from another author or copyright holder must demonstrate or obtain permission to publish this material under the Creative Commons Attribution 4.0 International (CC BY 4.0) License used by PLOS journals. Please closely review the details of PLOSu2019s copyright requirements here: PLOS Licenses and Copyright. If you need to request permissions from a copyright holder, you may use PLOS's Copyright Content Permission form.

Potential Copyright Issues:

i) Figure Figures 5 and 6. Please (a) provide a direct link to the base layer of the map (i.e., the country or region border shape) and ensure this is also included in the figure legend; and (b) provide a link to the terms of use / license information for the base layer image or shapefile. We cannot publish proprietary or copyrighted maps (e.g. Google Maps, Mapquest) and the terms of use for your map base layer must be compatible with our CC BY 4.0 license.

**Reviewers' comments:**

**Key Review Criteria Required for Acceptance?**

**Methods**

-Are the objectives of the study clearly articulated with a clear testable hypothesis stated?

-Is the study design appropriate to address the stated objectives?

-Is the population clearly described and appropriate for the hypothesis being tested?

-Is the sample size sufficient to ensure adequate power to address the hypothesis being tested?

-Were correct statistical analysis used to support conclusions?

-Are there concerns about ethical or regulatory requirements being met?

Reviewer #1: Yes, all methodos are ok

Reviewer #2: -The objectives are stated clearly in both the abstract and the introduction.

-Yes, the study design is appropriate to address the stated objectives.

-Yes, the population is clearly described and appropriate for the hypothesis being tested in the context of a systematic review.

- Yes, the sample size is sufficient to ensure adequate power to address the hypothesis being tested in this systematic review and descriptive analysis. This is the largest pooled sample to date for evaluating the global sociodemographic and clinical profile of mycetoma.

-Yes, appropriate statistical methods were used to support the conclusions, given the nature and aim of this systematic review and descriptive epidemiological synthesis. No interventional or comparative hypotheses were tested, so inferential statistics (e.g., p-values, confidence intervals, regression models) were not required.

-There are no major concerns about ethical or regulatory requirements being met in this study

Reviewer #3: The methodology of the study follows the recommended steps in a systematic review, followingISMA guideline with PROSPERO registration which indicate strong methodology

**Results**

-Does the analysis presented match the analysis plan?

-Are the results clearly and completely presented?

-Are the figures (Tables, Images) of sufficient quality for clarity?

Reviewer #1: Yes, the results are adequate.

Reviewer #2: -Yes, the analysis presented matches the implied analysis plan for a systematic review, based on the study's objectives and methods section.

-Yes, the results are clearly and completely presented, but there is some room for improvement in clarity and data visualization.

-Yes for the figures and tables

Reviewer #3: The data is well presented, and the figures and tables are high quality.

**Conclusions**

-Are the conclusions supported by the data presented?

-Are the limitations of analysis clearly described?

-Do the authors discuss how these data can be helpful to advance our understanding of the topic under study?

-Is public health relevance addressed?

Reviewer #1: Yes, the conclusions are adequate.

Reviewer #2: -The conclusions accurately reflect the scope, limitations, and strength of the findings. They are grounded in a comprehensive analysis of 72 studies and over 29,000 patients, and appropriately cautious where data was limited.

-Overall, the study's discussion of the data shows how it advances the understanding of mycetoma's epidemiology, clinical presentation, and geographic distribution, while also highlighting the need for further research, better data reporting, and improved diagnostic and treatment practices. This aligns with the goal of ultimately reducing the disease burden and improving outcomes for affected individuals.

-Yes, the authors discuss how their data can contribute to advancing our understanding of mycetoma

-the study directly addresses public health relevance by providing insights into disease prevention, treatment, surveillance, and data standardization. The authors propose actionable recommendations to improve public health strategies, from targeted interventions for at-risk populations to broader guidelines for global surveillance and research.

Reviewer #3: The conclusions are supported by the data presented, and the study limitations are clearly described.

**Editorial and Data Presentation Modifications?**

Reviewer #1: Accept

Reviewer #2: (No Response)

Reviewer #3: (No Response)

**Summary and General Comments**

Reviewer #1: This is a well-conducted, structured, and well-analyzed work. I practically accept it as is. The only suggestion I make is that the authors add a sentence in the discussion section indicating that some of the etiological agents were reported under previous names. For example, Nocardia asteroides is now a complex involving many species, and N. asiteroides sensu stricto is the smallest. In the Americas, the majority of cases are caused by M. psudomycetomatis, not M. mycetomatis as in Africa, and Trematosphaeria grisea could be Nigrograna mackinonii.

Reviewer #2: (No Response)

Reviewer #3: This article outstanding in its discipline since it is the first research to target the Epidemiological Profiling of Patients with mycetoma? It is also the most extensive systematic review on this topic. The methodology follows the recommended steps in the systematic review. The result is well presented and discussed in relation to the literature. The conclusion was well written, covering the objectives of the study. The limitation of the study was well described.

PLOS authors have the option to publish the peer review history of their article (what does this mean? ). If published, this will include your full peer review and any attached files.

**Do you want your identity to be public for this peer review?** For information about this choice, including consent withdrawal, please see our Privacy Policy .

Reviewer #1: **Yes: ** Alexandro Bonifaz

Reviewer #2: No

Reviewer #3: **Yes: ** MOHAMMED YOUSOF BAKHIET

**Figure resubmission:****Reproducibility:** To enhance the reproducibility of your results, we recommend that authors of applicable studies deposit laboratory protocols in protocols.io, where a protocol can be assigned its own identifier (DOI) such that it can be cited independently in the future. Additionally, PLOS ONE offers an option to publish peer-reviewed clinical study protocols. Read more information on sharing protocols at https://plos.org/protocols?utm_medium=editorial-email&utm_source=authorletters&utm_campaign=protocols

---

## [Editor Report · Decision Letter 1]

6 Jun 2025

Dear Dr Salah,

We are pleased to inform you that your manuscript 'Global Sociodemographic, Clinical, and Epidemiological Profiling of Patients with Mycetoma: A systematic review' has been provisionally accepted for publication in PLOS Neglected Tropical Diseases.

Best regards,

Max Carlos Ramírez-Soto, BSc, MPH, PhD, FRSPH, FECMM

Academic Editor

https://orcid.org/0000-0003-0471-6746

Joshua Nosanchuk

Section Editor

Shaden Kamhawi

co-Editor-in-Chief

Paul Brindley

co-Editor-in-Chief

---

## [Editor Report · Acceptance letter]

Dear Dr Salah,

We are delighted to inform you that your manuscript, "Global Sociodemographic, Clinical, and Epidemiological Profiling of Patients with Mycetoma: A Systematic Review," has been formally accepted for publication in PLOS Neglected Tropical Diseases.

Best regards,

Shaden Kamhawi

co-Editor-in-Chief

Paul Brindley

co-Editor-in-Chief
